# Viral Metagenomic Data Analyses of Five New World Bat Species from Argentina: Identification of 35 Novel DNA Viruses

**DOI:** 10.3390/microorganisms10020266

**Published:** 2022-01-24

**Authors:** Elisa M. Bolatti, Gastón Viarengo, Tomaz M. Zorec, Agustina Cerri, María E. Montani, Lea Hosnjak, Pablo E. Casal, Eugenia Bortolotto, Violeta Di Domenica, Diego Chouhy, María Belén Allasia, Rubén M. Barquez, Mario Poljak, Adriana A. Giri

**Affiliations:** 1Grupo Virología Humana, Instituto de Biología Molecular y Celular de Rosario (CONICET), Suipacha 590, Rosario 2000, Argentina; bolatti@ibr-conicet.gov.ar (E.M.B.); agustina.cerri@gmail.com (A.C.); dchouhy@detxmol.com.ar (D.C.); 2Área Virología, Facultad de Ciencias Bioquímicas y Farmacéuticas, Universidad Nacional de Rosario, Suipacha 531, Rosario 2000, Argentina; pablocasal380@gmail.com; 3DETx MOL S.A., Centro Científico Tecnológico CONICET Rosario, Ocampo y Esmeralda, Rosario 2000, Argentina; gviarengo@detxmol.com.ar; 4Institute of Microbiology and Immunology, Faculty of Medicine, University of Ljubljana, Zaloška 4, SI-1000 Ljubljana, Slovenia; Tomaz-mark.zorec@mf.uni-lj.si (T.M.Z.); lea.hosnjak@mf.uni-lj.si (L.H.); 5Museo Provincial de Ciencias Naturales “Dr. Ángel Gallardo”, San Lorenzo 1949, Rosario 2000, Argentina; euge_montani22@hotmail.com; 6Programa de Conservación de los Murciélagos de Argentina, Miguel Lillo 251, San Miguel de Tucumán 4000, Argentina; vdidomenica@live.com (V.D.D.); rubenbarquez@gmail.com (R.M.B.); 7Instituto PIDBA (Programa de Investigaciones de Biodiversidad Argentina), Facultad de Ciencias Naturales e Instituto Miguel Lillo, Universidad Nacional de Tucumán, Miguel Lillo 205, San Miguel de Tucumán 4000, Argentina; 8Área Estadística y Procesamiento de Datos, Facultad de Ciencias Bioquímicas y Farmacéuticas, Universidad Nacional de Rosario, Suipacha 531, Rosario 2000, Argentina; bortolotto.eugenia@gmail.com (E.B.); mallasia@fbioyf.unr.edu.ar (M.B.A.)

**Keywords:** *Chiroptera*, metagenomics, virome, virus identification, *Cressdnaviricota*, Cossaviricota, Anelloviridae

## Abstract

Bats are natural reservoirs of a variety of zoonotic viruses, many of which cause severe human diseases. Characterizing viruses of bats inhabiting different geographical regions is important for understanding their viral diversity and for detecting viral spillovers between animal species. Herein, the diversity of DNA viruses of five arthropodophagous bat species from Argentina was investigated using metagenomics. Fecal samples of 29 individuals from five species (*Tadarida brasiliensis, Molossus molossus, Eumops bonariensis, Eumops patagonicus*, and *Eptesicus diminutus*) living at two different geographical locations, were investigated. Enriched viral DNA was sequenced using Illumina MiSeq, and the reads were trimmed and filtered using several bioinformatic approaches. The resulting nucleotide sequences were subjected to viral taxonomic classification. In total, 4,520,370 read pairs were sequestered by sequencing, and 21.1% of them mapped to viral taxa. *Circoviridae* and *Genomoviridae* were the most prevalent among vertebrate viral families in all bat species included in this study. Samples from the *T. brasiliensis* colony exhibited lower viral diversity than samples from other species of New World bats. We characterized 35 complete genome sequences of novel viruses. These findings provide new insights into the global diversity of bat viruses in poorly studied species, contributing to prevention of emerging zoonotic diseases and to conservation policies for endangered species.

## 1. Introduction

Bats are natural reservoirs of a large variety of viruses, including many important zoonotic viruses causing severe diseases in humans and domestic animals, such as severe acute respiratory syndrome coronavirus (SARS-CoV), Ebola virus, Nipah virus, and Hendra virus [1,2,3,4]. In addition, because SARS-CoV-2 [5] and other coronaviruses [2,6] could have jumped to humans from bats through other intermediate hosts [5], scientific interest in these mammals has notably increased in recent years. The cross-species spillover events highlight the need for further characterization of bat viruses inhabiting different geographic regions in order to identify those with increased risk of cross-species transmissions. According to the database of bat-associated viruses (DbatVir, http://www.mgc.ac.cn/DBatVir/, accessed on 11 November 2021) [7], 13,059 viruses have been identified in bats globally, of which 1144 (8.8%) originated from South America, including 126 (0.96%) identified in Argentina. A total of 98.4% (124/126) of South American bat viruses belong to the family *Rhabdoviridae* (RNA viruses) and were identified by conventional molecular methods, mainly in arthropodophagous bat species during national rabies surveillance programs [8,9,10]. To date, only two DNA viruses have been identified in bats from Argentina, Tadarida brasiliensis papillomavirus type 1 (TbraPV1, *Papillomaviridae*) and Tadarida brasiliensis gemykibivirus 1 (*Genomoviridae*) [11].

Viral surveillance of wild bat populations is also necessary for proactive conservation management of this diverse taxonomic group. Specifically, viral shedding may serve as an indicator of broader environmental changes that lead to stress in the host population [12], inducing behavioral and physiological changes, and thus influence the disease dynamics among bats [13,14]. Because most bat-borne viruses are transmitted to other hosts by four routes—aerosols, droplets, fecal–oral contact, and direct contact—it is particularly important to determine the viral communities present in the gastrointestinal tracts of these animals.

Next-generation sequencing (NGS) technology combined with metagenomic analyses of the obtained nucleotide sequences constitute a powerful tool that has made it possible to identify an exponentially growing number of novel and emerging viruses in almost all types of clinical and environmental samples [15,16]. Therefore, to understand the true extent of viral genomic diversity, its origins, and its driving forces, the International Committee on Taxonomy of Viruses (ICTV, https://talk.ictvonline.org/, accessed on 25 September 2021) has been working since 2017 on a dynamic classification framework, which takes into account the current view that viruses have multiple origins (polyphyly) and that their diversity cannot be represented by a single virosphere-wide tree [15,17]. Thus, to expand our knowledge of viruses beyond the original parasite–pathogen model, and to identify historical events that played a crucial role in viruses’ origin and evolution, efforts are needed to provide novel viral taxa, especially from underrepresented species.

Using NGS, we recently described the oral/anal virome composition of *Tadarida brasiliensis* (I. Geoffroy Saint-Hilaire, 1824)*,* an arthropodophagous bat species widely dispersed across the Americas [11]. To expand the virome composition of *T. brasiliensis* from other transmission routes and to strengthen current knowledge about viral diversity in other New World bats (*Molossus* (Pallas, 1766)*, Eumops bonariensis* (Peters, 1874)*, Eumops patagonicus* (Thomas, 1924), and *Eptesicus diminutus* (Osgood, 1915)), this study applied a metagenomic approach to fecal samples of five bat species from Argentina. All bat species included in this study are arthropodophages and inhabit semiurban or highly urbanized areas, showing wide distribution in Argentina and in several regions of South America [18,19,20]. Our results provide novel insights into the viromes of different arthropodophagous bat species living in close contact with humans and contribute to knowledge of their possible role as pathogen reservoirs, providing key data for conservation purposes.

## 2. Materials and Methods

### 2.1. Study Area, Sample Collection, and Ethics Statement

Bat fecal samples were collected from two different geographical locations: the *T. brasiliensis* colony in downtown Rosario, Santa Fe, Argentina, as described previously [11], and other New World bat species (*M. molossus, E. bonariensis, E. patagonicus*, and *E. diminutus*) from Villarino Park in Zavalla, Santa Fe, Argentina (33°01′ S 60°53′ W), an extensive forested area close to urban settlements that hosts at least eight bat species.

In addition to the fecal samples of bats from the Rosario colony captured in our previous study [11], 53 individuals were captured using mist nets in Villarino Park from February to April 2017. Specifically, the bats were kept in individual cotton bags for species determination based on anatomical and morphological characteristics, reproductive status, and age [21]. Subsequently, fecal drops were collected from individual bags using sterile cotton-tipped swabs, suspended in 1 mL of viral transport media (VTM), and stored at 4 °C or on dry ice until further processing.

During this study, every effort was made to minimize animal disturbance and suffering; no breeding or pregnant females were captured, and no animals were harmed or required euthanasia. Sampling was carried out by trained professionals as approved by the Ministry of Environment of the Argentinian Province of Santa Fe (File 519/17) and the Animal Ethics Committee of the Faculty of Pharmaceutical and Biochemical Sciences (National University of Rosario, Rosario, Argentina, File 6060/243).

### 2.2. Sample Processing and Viral DNA Enrichment

Selected fecal samples from 29 individual bats were vortexed to completely resuspend the fecal material into the solution. Subsequently, 1 mL of Hank’s Balanced Salt Solution (HBSS) was added to each sample and further vortexed to create a less viscous solution. Suspensions were centrifuged at 10,000× *g* for 2 min and supernatants were transferred to fresh tubes and then pooled by species, sex, and collection site (Table 1). Following this criterion, six pooled samples were selected and prepared; each pool was filtered through a 0.45 µm pore size syringe filter (Fisher Scientific, Pittsburgh, PA, USA). The filtered supernatants were then centrifuged at 50,000× *g* for 3 h at 10 °C. Finally, each pellet was resuspended in 100 µL of HBSS and stored at −80 °C until further analyses.

To reduce the amount of contaminating RNA and DNA, each sample (116 µL) was treated with 14 U of DNase Turbo (Ambion, Austin, TX, USA), 25 U of Benzonase Nuclease (Novagen, Darmstadt, Germany), and 20 U of RNase One (Promega, Madison, WI, USA). Next, samples were diluted to a final volume of 140 µL in 10× DNase buffer (Ambion), incubated at 37 °C for 2 h, and immediately processed with the QiaAmp DNA Mini Kit (Qiagen, Valencia, CA, USA) using the manufacturer’s protocol to extract viral DNA that was protected from nuclease digestion by the viral capsid. Viral DNA was eluted to a final volume of 60 µL and then enriched using whole genome amplification (WGA) with the illustra Genomiphi V2 DNA Amplification Kit (GE Healthcare, Chicago, IL, USA).

### 2.3. Next-Generation Sequencing and Metagenomic Data Analyses

Indexed paired-end libraries from enriched pooled samples were prepared using the Nextera DNA Flex Library kit (Illumina, San Diego, CA, USA). Sequencing libraries were quantified using the Qubit 3.0 fluorometer (Thermo Scientific, Waltham, MA, USA), and DNA fragment size distribution was analyzed using the 2100 Bioanalyzer Instrument (Agilent, Santa Clara, CA, USA). Sequencing was performed on a MiSeq instrument (Illumina) with 600 cycles per sequencing read-pair (2 × 300 bp).

Reads were subjected to quality trimming and filtering using the bbduk program (BBTools v38.42), as described previously [11]. Host and human reads were subtracted by mapping to six bat reference genomes (https://bat1k.com/, accessed on 25 September 2021) and the human reference genome hg38, respectively, using Bowtie2 v2.2.4 [22]. Bacterial reads were subtracted by mapping the dataset to bacterial reference-index files (obtained 31 July 2021 from ftp://ftp.ccb.jhu.edu/pub/infphilo/centrifuge/data/p_compressed.tar.gz) using the Centrifuge sequence classification system (Centrifuge version 1.0.3-beta) [23].

De novo nucleotide sequence assembly was performed with SPAdes v3.15.3, using the default and meta pipeline parameter options, and MEGAHIT v1.2.9, setting default parameters [24]. Assembled contigs longer than 500 nt were filtered using CheckV v0.8.1 [25] and analyzed further.

Viral taxonomic classification of the cleaned reads and filtered de novo assembled contigs was performed using Centrifuge with the viral reference sequence database downloaded from NCBI RefSeq (obtained June 2021; Search string: txid10239 [Organism:exp] and “complete genome”) [23]. The results of viral taxonomic classification were further summarized to the taxonomic level of family using Pavian [26].

### 2.4. Characterization and Classification of Novel Metagenome-Assembled Viral Genome Sequences

Complete, nearly complete, and seemingly complete sequences exhibiting sequence similarity to known sequences of viral genomes, potentially representing previously uncharacterized viral sequences, were considered metagenome-assembled viral genomes (MAVGs) and were analyzed manually as follows. Sequence orientation, functional annotation, and, in cases of circular viruses, *ori* positioning were performed to best accommodate the general structural genomic characteristics of the taxonomic family or genus as indicated by the initial centrifuge and blast-based similarity/homology searches. These features were additionally revised after the final taxonomical classification had been established. Coding sequence domains were predicted as open reading frames (ORFs) exhibiting certain characteristics such as nucleotide/amino acid sequence similarity to known viral genes in the GenBank database (https://www.ncbi.nlm.nih.gov/genbank/, accessed on 25 September 2021) and/or containment of known functional sequence motifs.

The final taxonomic classifications were assigned according to the contemporary taxonomic guidelines of ICTV and taxonomic updates that had been recently published and were not yet included in the official ICTV taxonomy, as in the case of *Genomoviridae* [27], *Smacoviridae* [28], and *Anelloviridae* [29]. In addition, novel papillomavirus (PV) nucleotide sequences were subsequently submitted to the Animal Papillomavirus Reference Center (http://www.animalpv.org/, accessed on 25 September 2021) [30] for their confirmation and official designation.

The software packages Ugene (v40.0, Unipro) [31] and SnapGene Viewer 5.0.6 (Insightful Science, San Diego, CA, USA) were used to support the analyses. Sequence motifs were searched for in the nucleotide (nt) and amino acid (aa) sequences using general expression patterns. Pairwise sequence identities were calculated where indicated using a modified version of the sequence demarcation toolkit (SDT v1.0 for Linux 64bit) [32], using mafft (v7.4) [33] as the sequence aligner. The modification of SDT consisted of two steps preceding the actual pairwise sequence alignment, ensuring that both sequences were oriented in the same direction (using mafft v7.4) and, in the case of circular genomes, optimized the *ori* position using MARS [34]. Coverage statistics of novel MAVGs were estimated by remapping the trimmed read datasets to the MAVG sequences using Bowtie2 (v2.2.6) [22].

Phylogenetic analyses were performed out using iqtree (v1.6.12) [35] and, where indicated, model selection was performed using the built-in ModelFinder function [36]. Branch support was estimated as ultrafast bootstrap support values [37].

Phylogenetic analyses of potentially novel genomoviruses and smacoviruses, and novel viruses of the family *Circoviridae*, were based on the Rep protein multiple sequence alignments. The amino acid multiple sequence alignment of the Rep40 proteins was used to phylogenetically position the potentially novel parvoviruses, whereas a multiple nucleotide sequence alignment of the *ORF1* gene was used for the phylogenetic analysis of the potentially novel anelloviruses. The context sequences used for phylogenetic analyses of the potentially novel genomoviruses (n_context sequences_ = 109), smacovirus (n_context sequences_ = 84), and anelloviruses (n_context sequences_ = 987) were selected based on the most recent relevant taxonomical updates [24,25,26] and downloaded from GenBank. Context sequences for phylogenetic reconstruction and placement of the potentially novel parvovirus (n_context sequences_ = 56) and viruses belonging to the family *Circoviridae* (n_context sequences_ = 90) were downloaded from the relevant ICTV resource pages (https://talk.ictvonline.org/ictv-reports/ictv_online_report/ssdna-viruses/w/parvoviridae/1055/resources-parvoviridae, accessed on 1 September 2021 and https://talk.ictvonline.org/cfs-file/__key/communityserver-wikis-components-files/00-00-00-00-83/OSD.Cir.Fig1A.Cyclovirus_5F00_circovirus_5F00_reps_5F00_aln_5F00_ed.fas, accessed on 1 September 2021). In addition, in the case of *Circoviridae*, the first three most similar RefSeq sequences according to blastn searches not yet present among the primary ICTV sequences were added to the database of context sequences.

Phylogenetic analysis of potentially novel PVs identified in this study was performed as described previously [11], using the concatenation of the *E1, E2, L2,* and *L1* gene sequences of 384 reference PV genomes (downloaded 27 September 2021 from PaVe http://pave.niaid.nih.gov/, accessed on 15 October 2021) [38] and the corresponding genes from potentially novel PVs.

### 2.5. Diversity Analyses

To analyze viral diversity of the individual bat species and to compare them among the six sample pools, diversity indexes were calculated based on the proportions of reads mapping to the contigs representing individual viral families or unclassified viral nucleotide sequences. The diversity indices were estimated as Rényi entropies with α = 0, 0.25, 0.50, 0.75, 1 and 2. To quantify the compositional dissimilarity between specific sample pools, β-diversity was estimated by calculating intersample Bray–Curtis dissimilarities. All analyses were performed using the R package Vegan [39,40].

### 2.6. Nucleotide Sequence Accession Numbers

The novel viruses reported in this article are openly available in the GenBank/EMBL/DDBJ database with the following accession numbers: OL704823–OL704858. The relevant raw high throughput sequencing data obtained in this study were deposited at the NCBI Sequence Read Archives (SRA) under BioProject ID PRJNA786972.

## 3. Results

### 3.1. High-Throughput Sequencing Data Analyses

A total of 29 samples of bat feces, grouped into six sample pools (Table 1 and Appendix A), were included in the metagenomic analyses. As shown in Table 2, a total of 4,520,370 sequencing read pairs were obtained, of which 3,750,093 passed the procedures/criteria for quality trimming and filtering. An additional 26,418 and 293,007 read pairs were excluded from further analyses because of their mammalian and bacterial/archaeal origin, respectively. The remaining 3,340,668 read pairs were submitted to three different de novo assemblers, which together yielded 17,663 contigs longer than 500 nt (Table 2).

In total, 724,189 read pairs (21.1% of all pairs that passed quality filtering and host/bacteria subtraction) and 691 assembled contigs (3.91% of all contigs) mapped to viral taxa (Table 2), corresponding to 41 viral families (Figure 1A, Appendix A). Most viral read pairs were related to families of animal-infecting viruses (98.37% of viral read pairs), followed by families of viruses infecting bacteria or archaea (1.45%), protists (0.089%), and plants (0.005%; Figure 1B, Appendix A). On the other hand, collectively, most viral contigs obtained from de novo assemblies represented viral families infecting bacteria and archaea (59.9% of viral contigs), followed by viruses infecting animals (35.0% of viral contigs), protists (1.59% of viral contigs), and plants (0.434% of viral contigs). In addition, 21 viral contigs (3.04%), most of which were circular Rep-encoding single-stranded (CRESS) DNA viruses, were found related to so far unclassified viruses without a formally determined host tropism (Appendix A).

The most frequent viral families infecting animals were *Circoviridae* and *Genomoviridae*, which were identified in all bat sample pools tested. The most commonly represented bacterial/archaeal viral families were *Microviridae*, *Myoviridae*, and *Siphoviridae*, likely reflecting the bacteria present in the bat digestive system. Similarly, nucleotide sequences related to plant-, arthropod-, and protist-infecting viral families likely represent the diet of the bats or the plant diet of the arthropods ingested by the bats and their parasites (Figure 1A and Appendix A).

Sample pools P1 and P2, originating from *T. brasiliensis*, stood out in terms of the highest viral read counts among all pools, but they yielded relatively lower numbers of viral contigs, which corresponded to their diminished viral diversity (Figure 1, Table 3 and Appendix A). In further detail, a large portion of viral read pairs (P1: 98.7% and P2: 99.9%) corresponded to the family *Circoviridae*. On the other hand, these reads reassembled into only five viral contigs, two of which, one from each of the two pools, indicated extremely high sequence-wide fold coverages (9,741.64× and 63,800× in P1 and P2, respectively; Figure 1, Table 2, Appendix A). The two sequence contigs were identical to each other, contained sequence features that suggested the near completeness of the encoded viral genome, and were analyzed in further detail as MAVG09 (P1) and MAVG10 (P2).

### 3.2. Comparative Analysis of Viral Diversity

Viral diversities within each sample pool were evaluated using Rényi’s entropy. A pronounced difference was observed among diversity index values of the two *T. brasiliensis* pools (P1 and P2) in comparison to those obtained from other bat species included herein. Significantly lower α-diversity values in P1 and P2 suggested a lower viral diversity among bats of the Rosario *T. brasiliensis* colony (Table 3 and Appendix A). On the other hand, our results did not suggest differences in α-diversities between sample pools obtained from bats inhabiting Villarino Park (Table 3 and Appendix A).

To facilitate compositional diversity comparisons, as estimates of β-diversity, pairwise Bray–Curtis dissimilarities (*D_BC_*) were calculated (Table 4). Overall, the highest dissimilarity, 0.999, was observed between P2 and P11, and the lowest, 0.125, between P4 and P9. Clearly, two cliques of sample pools formed based on the compositional dissimilarities. Sample pools P1 and P2, both from the Rosario *T. brasiliensis* colony, were more similar to each other than to any other sample pool (0.684); in a similar manner, sample pools obtained from Villarino Park (P4, P6, P9, P11) were more similar to each other than to sample pools from the *T. brasiliensis* colony, and to a much higher degree, although they were collected from different species of bats. The highest dissimilarity between sample pools from Villarino Park was observed between P6 and P11 (0.461).

### 3.3. Identification of Novel Bat-Associated Viruses

Of the reconstructed viral contigs, 35 represented complete or nearly complete viral nucleotide genome sequences, MAVGs, that were used in further analyses, as shown in Table 5. In addition, genomic features, annotations, and coverage of the 36 MAVGs identified are provided in Appendix A, and generic genome maps of the six viral families identified herein are depicted in Appendix A. Most MAVGs identified in this study belonged to viral families of the recently established taxonomic phylum Cressdnaviricota [17,41] (n = 30), including *Genomoviridae* (*n* = 18), *Circoviridae* (*n* = 11), and *Smacoviridae* (*n* = 1). In addition, four MAVGs belonged to the phylum *Cossaviricota*, with three MAVGs corresponding to the family *Papillomaviridae* and one MAVG corresponding to *Parvoviridae*. Finally, two MAVGs belonged to the family *Anelloviridae* (Table 5, Appendix A).

#### 3.3.1. Genomoviridae

A total of 18 MAVGs belonged to the family *Genomoviridae*. Most of them were from sample pools of *M. molossus* (13/18; two viruses in sample pool P6, 11 viruses in P9), followed by *E. patagonicus* or *E. diminutus* (3/18; P4) and *E. bonariensis* (2/18; P11; Table 5). On the basis of the Rep amino acid sequence phylogeny, the novel genomoviruses belonged to the genera *Gemycircular-* (MAVGs13, 14, 15, 19, 23 and 25), *Gemykibi-* (MAVGs12, 16, 17, 18, 21, 22 and 24), *Gemykronza-* (MAVG26, MAVG29), and *Gemygorvirus* (MAVG20) (Figure 2). With the exception of MAVG15, all other genomoviral MAVGs represented novel species, according to the suggested species demarcation threshold [27]. The MAVGs 12 and 24 represented the same viral species of the genus *Gemycircularvirus*. The MAVG15 exhibited high complete genome sequence identity (88.4%) to the *Genomoviridae* sp. isolate wftbif32cir1 (GenBank acc. no. MT138090), originally identified in a bird metagenome sample from China. The two viruses belong to the same species of *Gemycircularvirus*; however, the species has not yet been recognized. We were unable to locate the *Genomoviridae*- and *Geminiviridae*-specific GRS motifs in MAVG12; furthermore, the genome indicated unusual positioning of the RCR motif III, which appeared past Walker motif A, well into the SF3 helicase domain of the amino acid sequence. Interestingly, the sequence MAVG20 contained two C-terminal Walker C motifs, both sharing an identical amino acid sequence pattern WLTN (Appendix A). The two WLTN amino acid motifs also shared an identical nucleotide sequence, suggesting that the duplication may be due to an assembly error.

#### 3.3.2. Circoviridae

Of the 11 MAVGs (MAVG01–10, MAVG30) representing novel sequences of viruses from the family *Circoviridae*, three (MAVG01–03) and seven (MAVG04–10) MAVGs were tentatively assigned to the genera *Cyclovirus* and *Circovirus*, respectively. On the other hand, MAVG30 was not assigned to either of the two genera because of its unique genomic characteristics (Table 5 and Appendix A, Figure 3).

Peak complete sequence identities between the *Circoviridae* MAVGs and extant complete genome sequences of other representatives of *Circoviridae* (obtained from ICTV and RefSeq, taxID: 39224) ranged from 49.3 to 74.5%. MAVG05 and MAVG06, otherwise most similar to complete genome sequences of Canine circovirus (GenBank acc. no. JQ821392) and Feline cyclovirus (GenBank acc. no. KM017740), respectively, were 83.8% identical to each other. Similarly, MAVG09 and MAVG10 were perfectly identical to each other and exhibited peak complete sequence similarity to the genome sequence of Culex circo-like virus (GenBank acc. no. NC_040567). Based on current ICTV criteria for species demarcation in the family *Circoviridae,* all of these MAVGs represented novel species of *Circoviridae.* MAVG09 and MAVG10 represented the same virus, and MAVG05 and MAVG06 represented two different types of the novel same viral species.

Notably, MAVGs 04 and 05 presented unusual configurations of sequence motifs, exhibiting the apex nonanucleotide motifs in reverse complement. In fact, strictly following the ICTV guidelines (genus assignment based on the orientations and positionings of the *Rep* and *Cap* genes in relation to the position and direction of the stem loop apex nonanucleotide motif) [42], these two MAVGs were initially classified as cycloviruses, and the classification was reconsidered only upon reviewing the Rep amino acid phylogenetic tree (Figure 3), which placed them immediately adjacent to MAVGs 06 and 07, which were embedded within the genus *Circovirus*.

Because the two major coding sequences, *Rep* and *Cap*, which are encoded on the opposite strands of their ssDNA in cycloviruses and circoviruses [42], were located on the same, most likely positive, strand in the case of MAVG30, MAVG30 could not be assigned to either of the two genera, as mentioned above. Moreover, in the Rep amino acid phylogenetic tree, MAVG30 was positioned alongside other *Circoviridae* viruses that had not been assigned to either genus (Figure 3; GenBank acc. nos. NC_025722, NC_030457) but gravitated toward the genus *Circovirus* rather than *Cyclovirus*. The functional motifs found in the helicase domain of the MAVG30 Rep protein sequence resembled those commonly found within the family *Circoviridae*, whereas the motifs in the RC endonuclease domain did not fit the patterns characteristic of any individual family listed in the ICTV proposal for the phylum *Cressdnaviricota* (ICTV proposal 2019.012D). Finally, in MAVG30, the RC endonuclease motifs II and III were identified by the general patterns xHxQx and YxxK, respectively, while the search pattern for motif I was compiled from the motif I patterns of the other *Cressdnaviricota* families.

#### 3.3.3. Papillomaviridae

The three *Papillomaviridae* MAVGs found in this study (MAGV32–34) were identified in *M. molossus* and *E. bonariensis* fecal samples and officially named Molossus molossus papillomavirus type 2 (MmoPV2), Eumops bonariensis papillomavirus type 1 (EbonPV1), and Eumops bonariensis papillomavirus type 2 (EbonPV2) [43] (Table 5). Pairwise comparison of 387 PV *L1* gene nucleotide sequences indicated that MmoPV2 and EbonPV2 shared 80.7 and 71.4% nucleotide identities with Molossus molossus papillomavirus type 1 GenBank acc. no.: KX812447), respectively, and EbonPV1 showed the highest *L1* gene identity with human papillomavirus (HPV) type 41 (GenBank acc. no.: NC_001354; 70.5%). According to our analyses, MmoPV1 and MmoPV2 are members of the same unclassified species. On the other hand, EbonPV2 shows a basal taxonomic position with respect to MmoPV1 and MmoPV2, representing a potentially separate species within an unclassified PV genus. In addition, EbonPV1 could therefore be a novel member of the genus *Nupapillomavirus*, possibly belonging to a novel unclassified species (Figure 4).

Detailed analysis of the three putative novel PVs (Appendix A) showed typical genomic organization of bat PVs, potentially encoding five early genes (*E6, E7, E1*, *E2*, and *E4*) and two late genes (*L2* and *L1*) [41,42]. No canonical *E4* ORF could be identified in the genome of MmoPV2; however, a short sequence of 111 amino acids (nt positions 3160–3465), with residues of the proline-rich stretches that characterize E4, was detected. The three putative PVs presented an upstream regulatory region (URR1) spanning between the stop codon of *L1* and the start codon of *E6*. In addition, MmoPV2 and EbonPV2 contained a second noncoding region (URR2) between the early and late regions, as described in other bat PVs [44,45,46].

#### 3.3.4. Anelloviridae

The MAVGs 35 and 36 belonged to the family *Anelloviridae*, and the viruses were named Torque teno Tadarida brasiliensis virus 2 and Torque teno Tadarida brasiliensis virus 3, respectively (Table 5). Phylogenetic analysis based on *ORF1* nucleotide sequences indicated that Torque teno Tadarida brasiliensis virus 2 could be the prototype of a novel *Anelloviridae* species within the genus *Thetatorquevirus*, whereas the phylogenetic analysis of Torque teno Tadarida brasiliensis virus 3 convincingly suggested that this virus may be the founding member of a novel *Anelloviridae* genus (Figure 5).

#### 3.3.5. Smacoviridae

One MAVG (MAVG11) from sample pool P9, originating from *M. molossus*, represented a novel virus of the taxonomic family *Smacoviridae* (Table 5). According to its placement in the Rep amino acid phylogenetic tree (Figure 6), the virus represents a novel viral species within the genus *Porprismacovirus*. The virus was named Molossus molossus associated porprismacovirus 1.

#### 3.3.6. Parvoviridae

MAVG31, identified in sample pool P1 from *T. brasiliensis*, was assigned to the taxonomic family *Parvoviridae*, and more specifically to the genus *Dependoparvovirus*. It exhibited sufficient sequence divergence from known parvovirus sequences to be considered a novel species [47]. The virus was named Tadarida brasiliensis associated dependoparvovirus (Figure 7).

MAVG31 very likely represents a nearly complete viral genome sequence, lacking both a 5′-terminal polyadenylation site and a 5’-terminal inverted repeat region characteristic of *Parvovirinae*. On the other hand, while its *NS1* does carry a sensible set of start/stop codons, comparison with other *Dependoparvovirus* NS1 sequences revealed that it may encode only one of the truncations of the viral Rep protein, Rep58 (Table 5 and Appendix A).

## 4. Discussion

With 21 families and 1,411 species distributed throughout the world, except in the polar regions, bats (order *Chiroptera*) are the second-largest group of mammals [48,49]. Several factors, such as their relatively long lifespan and their ability to fly, allow them to have a wider range than other terrestrial mammals and more frequent direct or indirect contacts with other animal species in various geographical locations [50]. On the other hand, because of various immunological and metabolic adaptations, bats appear to be adept controllers of viral infections [51,52] and have an unusually high capacity for hosting a wide variety of viruses, including some that have been found highly pathogenic to humans [1,2,3,4,5,6,53]. Many instances of viral spillovers in the past have been attributed to various ecological factors—stressors—such as local climate and habitat changes, and it has been speculated that these factors had influenced the viral abundance and virome dynamics in the bat populations [54,55].

This study evaluated and compared the virome composition of pooled fecal samples of five arthropodophagous South American bat species (*Molossidae*: *T. brasiliensis*, *M. molossus*, *E. bonariensis*, *E. patagonicus*; *Vespertilionidae*: *E. diminutus*). The samples were obtained from two ecologically distinct sites in Argentina: an urban location, downtown Rosario, and a rural location, Villarino Park in Zavalla, in the province of Santa Fe.

The samples were grouped into six sample pools according to bat species, with the exception of one sample pool (P4), which contained fecal samples from bats of two different species (*E. diminutus* and *E. patagonicus*). Using high-throughput sequencing and our metagenomic classification pipeline, we obtained a total of 725,148 virus-related read pairs (21.1% out of 3,430,668), which were subsequently assembled into 691 contigs (3.91% out of 17,663) that mapped to viral taxa. Collectively, the read pairs and assembled sequences from all sample pools represented 41 different viral families, most of which were viral families infecting animals (21 families), followed by bacterial and archaeal viruses (16 families) and plant- and protist-infecting viruses (two families each). Similar viral metagenomic studies from around the world obtained similar viral diversities; for example, a study interrogating the viromes of four New World bat species, including *T. brasiliensis,* from roosts in Northern California (USA) revealed the presence of 31 viral families [56]; in a study of 18 species of Old World bats from Switzerland, the authors observed 39 different viral families [57]; a Croatian study revealed 63 viral families in Old World bats of seven different species [58]; and an African study based in Guinea detected 10 viral families [59]. As already pointed out by others, because of several confounding factors, comparing results of different metagenomic studies is difficult to say the least because studies differ in interrogated sample types, pooling strategies, sample preparation, and bioinformatic postprocessing [57,58,60]. Unlike most bat viral metagenomic studies, we focused on DNA viruses. Our previous study, focusing only on the oral/anal swabs of the *T. brasiliensis* colony in Rosario, identified 43 different viral families [11], whereas our results in this study indicated only nine different viral families in samples obtained from bats of the same colony. As discussed above, our two studies differed in the types of samples used, because this study included only fecal samples. Previously, bacterial/archaeal viruses were most prevalent, whereas in this study, the results favored animal-infecting viruses. Furthermore, in this study, viral enrichment, achieved by ultracentrifugation and filtration of samples prior to nucleic acid extraction, was reflected in a much higher yield of viral read pairs and viral contig proportions: 21.1% and 3.91%, respectively, versus 0.534% and 0.357%, respectively, without enrichment in the previous study [11].

Although we focused specifically on DNA viruses, and there was no reverse transcription step in our sample preparation procedures, our results suggested the presence of six taxonomic families of RNA viruses: *Xinmoviridae*, *Retroviridae*, *Tobaniviridae*, *Rhabdoviridae*, *Flaviviridae*, and *Paramyxoviridae.* To some degree, detection of *Retroviridae* is not surprising, because many retroviruses have become integrated and naturalized into their host genomes [61]. On the other hand, phi29 has been previously reported to possess some degree of reverse transcriptase activity [62]. Finally, in comparison to our previous study [11], a more conservative filtering approach was used in this study in the downstream bioinformatic analysis: only the viral taxonomic entities that were represented by contigs obtained from de novo assemblies were considered valid. Thus, the families of the RNA viruses detected could be sequencing artifacts or artifacts originating from incorrect metagenomic classification, because both fewer viral families and fewer families of RNA viruses were suggested by the results of this study in comparison to our previous results [11]. Moreover, recent evidence has shown the identification of endogenous viral elements (EVEs) from positive ssRNA (i.e., *Flaviviridae*), negative ssRNA (i.e., *Rhabdoviridae*, *Xinmoviridae*), and different types of segmented RNA viruses integrated in eukaryotic genomes, including insect genomes [63]. It may be that the detected sequences from the other RNA viral families are also due to EVEs.

A total of 35 complete and nearly complete viral genome sequences of novel DNA viruses were identified and characterized during this study, clustering to several viral families: *Circoviridae*, *Genomoviridae*, *Papillomaviridae*, *Parvoviridae*, *Smacoviridae*, and *Anelloviridae*. They represented 34 different viral species, of which 31 were newly discovered by the results of this study, and 13 viral genera, with one potential founder of a novel *Anelloviridae* genus.

*Genomoviridae*, *Circoviridae*, and *Smacoviridae* are taxonomic families, recently unified into the common phylum *Cressdnaviricota*, containing circular Rep-encoding single-stranded DNA viruses (CRESS DNA viruses). Many new CRESS DNA viruses have been identified in recent years using metagenomic analyses of whole-genome shotgun sequencing experiments of various types of samples. Because of this, their true host/tissue tropism and status as pathogens has been difficult to assess [41,64]. The 18 novel genomoviruses identified in this study were phylogenetically placed into four different viral genera and founded 17 novel viral species. Since the identification of the first genomovirus identified by our group in a New World bat species —specifically, in oral/anal swabs of *T. brasiliensis* [11]— only one other genomovirus has been identified in any species of New World bats [65]. The novel smacovirus identified in this study is the first smacovirus found in association with bats, and it was classified as a novel species of the genus *Porprismacovirus*. To date, smacoviruses have been identified mainly in animal fecal samples, with no reports of these viruses in bats or any other animals from Argentina [28]. The novel viruses of the family *Circoviridae* that have been identified in this study, with the exception of three, were phylogenetically related to mammal-associated viruses, which may suggest that the bats may represent their actual hosts. A previous study suggested that circovirus sequences detected in mammals are phylogenetically more closely related to each other than to circoviruses found in avian hosts [66]. On the other hand, the viruses Tadarida brasiliensis associated circovirus 1, Bat associated cyclovirus 17, and Molossus molossus associated CRESSDNA virus were phylogenetically more closely related to insect-associated viruses than to mammal-associated viruses of the family *Circoviridae*; these may have been hosted by insects, which were part of the bats’ diet. Moreover, as suggested previously for the case of a bat herpesvirus [67], perhaps the presence of certain CRESS DNA viruses, as integral parts of the bat microbiome, could to some degree serve as a predictor of disease, cross-species transmissions, habitat loss, and so on and might provide valuable information for conservation policies of species protected by international treaties, such as those mentioned in this report.

To date, more than 600 PVs have been described, and at least half of them are HPVs (https://pave.niaid.nih.gov/#home, accessed on 25 September 2021) [38]. In particular, a total of 15 bat PVs have been identified, mostly in oral/anal samples of Old World bat species [44,45], with only four PVs reported in two South American bat species: *T. brasiliensis* (TbraPV1–3) and *M. molossus* (MmoPV1) [11,46]. Here we report EbonPV1 and EbonPV2 as the first PVs identified in *E. bonariensis*, and MmoPV2, which would be the first officially confirmed PV identified in *M. molossus*, because the previously reported MmoPV1 (GenBank acc. no. KX812447) [46] had a 1,019 bp deletion within the *E2* gene (complete genome alignment nt position: 3,422–4,440). Phylogenetically, EbonPV1 and EbonPV2 appeared distantly related to each other, sharing only 57.8% nucleotide identity along the *L1* gene. Moreover, EbonPV2 clustered into the same unclassified genus with MmoPV1 and MmoPV2, both identified in *M. molossus*, while EbonPV1 interestingly appeared to be closely related to HPV41, therefore representing a novel member of the genus *Nupapillomavirus*. It has been suggested that multiple evolutionary forces have influenced the evolution of bat PVs, including coevolution, adaptive radiation, broad host range, host switch, and recombination events, because they display a highly polyphyletic pattern throughout the *Papillomaviridae* phylogenetic tree [11,45]. It is worth noting that neither HPV41 nor any closely related HPVs have been identified in human samples since the identification of HPV41 more than 30 years ago [68,69], despite the hundreds of specimens that have been tested in numerous epidemiological and clinical studies [70,71]. The hypothesis that HPV41 could be an inter-species recombinant virus arises from the identification of a gene transfer event during PV evolution involving ancestors of a porcupine papillomavirus (EdPV1) and HPV41 [72]. In addition, a recent study demonstrated that HPV41 has an atypical ability to encode a miRNA that could be explained by its possible recombinant nature [73]. Multiple recombination events have been described during PV evolution, mostly involving the *E2*–*L2* region [74,75] and the *E1* ORF [76]. The presence of a second URR in the intergenic *E2*–*L2* region observed in distantly related PVs, such as the MmoPV2 and EbonPV2 identified herein, may therefore be the result of individual and independent recombination events during their evolution [45]. These observations reinforce the importance of characterizing PVs from understudied taxonomic groups, which may help elucidate the evolutionary history driving papillomavirus diversification and, therefore, the clinical implications of their infections.

This study detected two novel anelloviruses distantly related to each other in a sample pool from *T. brasiliensis.* Both viruses were only distantly related to the only two previously reported bat anelloviruses, Torque teno chiroptera virus 1 (*Xitorquevirus*) [77] and Torque teno Desmondus rotundus virus (*Sigmatorquevirus*) [78], phylogenetically grouping with completely different *Anelloviridae* genera. Torque teno Tadarida brasiliensis virus 2 clustered alongside members of the genus *Thetatorquevirus*, and Torque teno Tadarida brasiliensis virus 3 was positioned basal to the genus *Wawtorquevirus*, possibly seeding a novel genus in the *Anelloviridae* taxonomy. In recent years, and with the advent of metagenomic technology, anelloviruses have been detected in different types of samples, including blood, tissues, and feces of a broad spectrum of mammals [78,79]. Torque teno virus viraemia (and/or) shedding has been associated with many diseases in humans, including hepatitis, multiple sclerosis, and hepatocellular carcinomas, but there is no confirmation that anelloviruses are the etiological agents of any human diseases [80,81]. Notably, Torque teno virus viraemia has been proposed as a clinical indicator of excessive immune suppression in patients following liver and/or kidney transplantation [82]. On the other hand, anelloviruses have been associated with pathologies in multiple organ systems in pigs and chickens [83,84,85]. Thus, it is unclear whether the two novel bat anelloviruses are associated with any pathologies, because, based on their detection in fecal samples, they might represent normal viral flora of the bats’ intestines.

A nearly complete genome sequence of a novel bat dependoparvovirus, phylogenetically related to a dependoparovirus detected in bats from China, was identified in fecal samples of *T. brasiliensis*. According to previously published studies, the identification of novel parvoviruses in diverse samples from both vertebrate and invertebrate hosts has increased in recent years [47]. Moreover, a high diversity of parvoviruses, particularly in the genus *Dependoparvovirus*, has been found in samples from New and Old World bats, and although they have not been associated with diseases, it has been pointed out that bat parvoviruses may have the potential to cross the species barrier and infect new hosts [86,87,88]. Therefore, our findings reinforce previous reports demonstrating a remarkable diversity of bat parvoviruses [86,87] and provide novel data regarding the ecology and evolution of the family *Parvoviridae*.

Most strikingly, our results of the quantitative comparisons of virome composition showed a stronger correlation between virome composition and the location from which the sample pools were obtained than between virome composition and bat species. Consistently, all samples from the rural location, Villarino Park, had higher species richness and alpha diversity than the urban colony of *T. brasiliensis* from downtown Rosario, because almost all reads in these two sample pools supported the same genome sequence, representing the newly identified Tadarida brasiliensis associated circovirus 1. The influence of habitat on viral diversity has recently been proposed as one of the main influences on viral diversity in neotropical rodents, with the lowest viral richness observed in periurban areas [89]. In addition, previous studies have found that the richness of viral communities in fecal samples of New and Old World bat species decreases with local anthropogenic food resources [54,55]. In this context, it could be that the decline in viral diversity in the Rosario colony was a consequence of the urban environment, with its lower diversity of different insects, the diet of bats. In line with this interpretation, the virus Tadarida brasiliensis associated circovirus 1 may have actually been hosted by an insect rather than a bat. On the other hand, it seems interesting that this exact same viral genome was found in two different sample pools with extremely high sequence coverage in both, suggesting that individuals throughout the Rosario colony were shedding this one specific virus. Although bats are considered ideal hosts and adept controllers of viral infection [51,52], spillovers of zoonotic viruses to humans have often been reported as episodic, transient, and/or seasonal, associated with increased viral shedding due to environmental or reproductive stress [4,90,91,92,93,94]. Bat colonies have been observed to respond to stress by shedding viruses, such as Nipah and Hendra, to which exposure can be lethal to many mammals other than bats, including horses, pigs, and humans [4,90]. In addition, a high viral load of coronaviruses and astroviruses has been observed in a *Myotis* colony in association with the reproductive success of the colony [95]. Because the Rosario *T. brasiliensis* colony is a large and migratory maternity colony settled in a building [21,96], it could be that the infection with this virus had exacerbated and spread throughout the colony because of stress related to the bats’ reproductive cycle.

Although these hypotheses arising from our work are interesting, they require thorough confirmation. In particular, future studies should attempt to disentangle confounding factors influencing bat virome composition. Clearly, the virome composition of the same bat species at different locations with varying degrees of human activity should be studied systematically and with great rigor. On the other hand, and as shown by the results of this study, the temporal component should also be carefully investigated to account for seasonal and episodic dynamics.

## 5. Conclusions

The viral metagenomic data presented in this study provide a snapshot of the virome of some rural and urban New World bats. A total of 35 novel DNA viruses, clustering to six different viral families and associated with five different bat species, were identified. Our results suggest a correlation between viral diversity and sampling location, regardless of species. Future studies should improve sampling to include samples of the same species from different locations, as well as those obtained on different occasions at regular intervals, to distinguish temporal effects associated with bat virome dynamics. Further characterization of the bat virome will improve our understanding of mammalian viral diversity and timely detection of potential human pathogens. This study represents an important contribution to better understanding the global diversity of bat viruses in poorly studied species for current and future prevention of emerging zoonotic diseases, source-tracking, and prediction, as well as for conservation policies for endangered species.

## Figures and Tables

**Figure 1 microorganisms-10-00266-f001:**
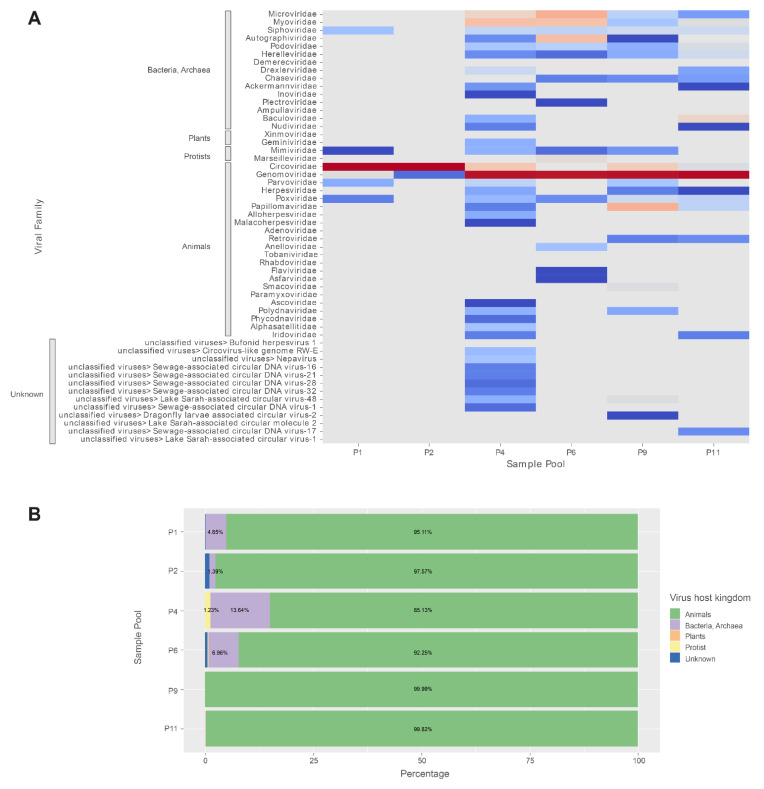
Abundance of viral families in pooled fecal samples of five bat species from Argentina identified by metagenomics. (**A**) Heatmap of relative abundances of viral families according to the number of reads found in each sample pool. The viral families are grouped according to host kingdom. (**B**) Relative abundance of viral reads classified at the kingdom level in each sample pool. Percentages higher than 1% are shown. P1 and P2: *T. brasiliensis*; P4: *E. diminutus* and *E. patagonicus*; P6 and P9: *M. molossus*; P11: *E. bonariensis*.

**Figure 2 microorganisms-10-00266-f002:**
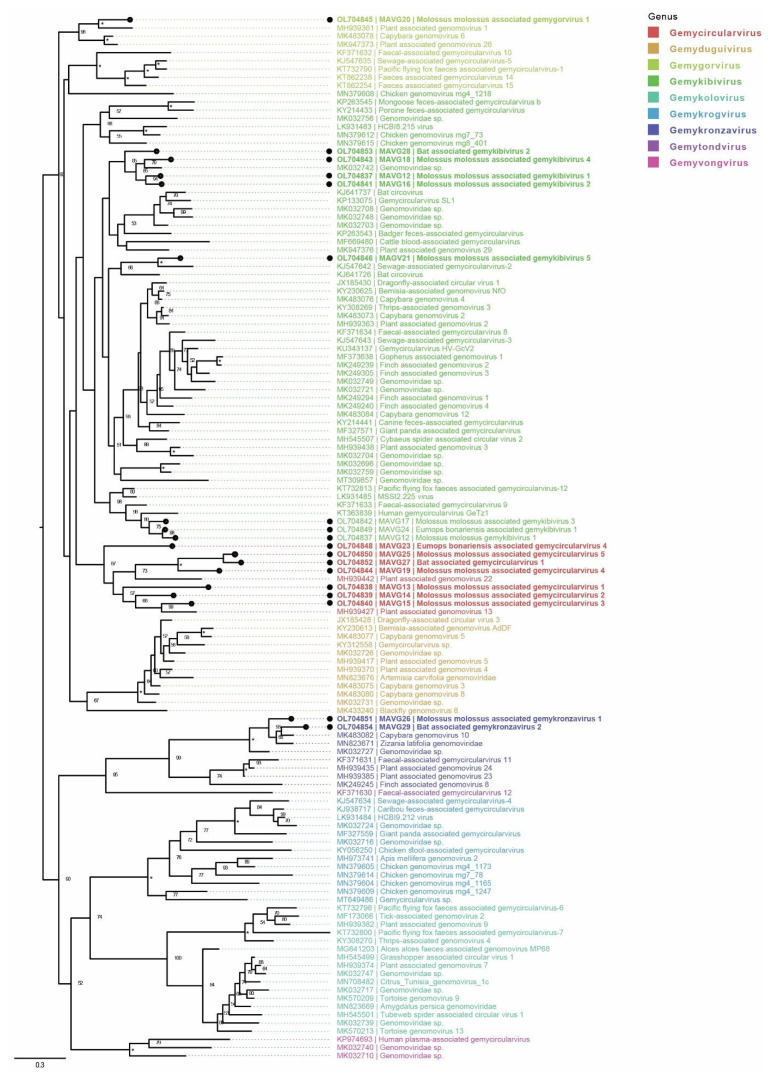
Placement of 18 novel genomoviruses into the phylogenetic context of other viruses clustering to the family *Genomoviridae*. The phylogenetic tree was built based on the Rep protein multiple sequence alignment (aa), which was produced using mafft (v7.45) [30]. The sequences evaluated in the most recent taxonomic update of the family *Genomoviridae* [27] were downloaded from GenBank and used as context (*n* = 109). Newly identified viruses are marked with black dots, and genera are color-coded. Saturated node support values are shown with asterisks (*). The phylogenetic tree was constructed using iqtree v1.6 [35] with 1,000 UFBootstrap replicates [37] and the phylogenetic model LG+F+R9. Tree visualization was facilitated using Figtree v1.4.4 (https://github.com/rambaut/figtree.git, accessed on 25 September 2021), and the tree was rooted at midpoint.

**Figure 3 microorganisms-10-00266-f003:**
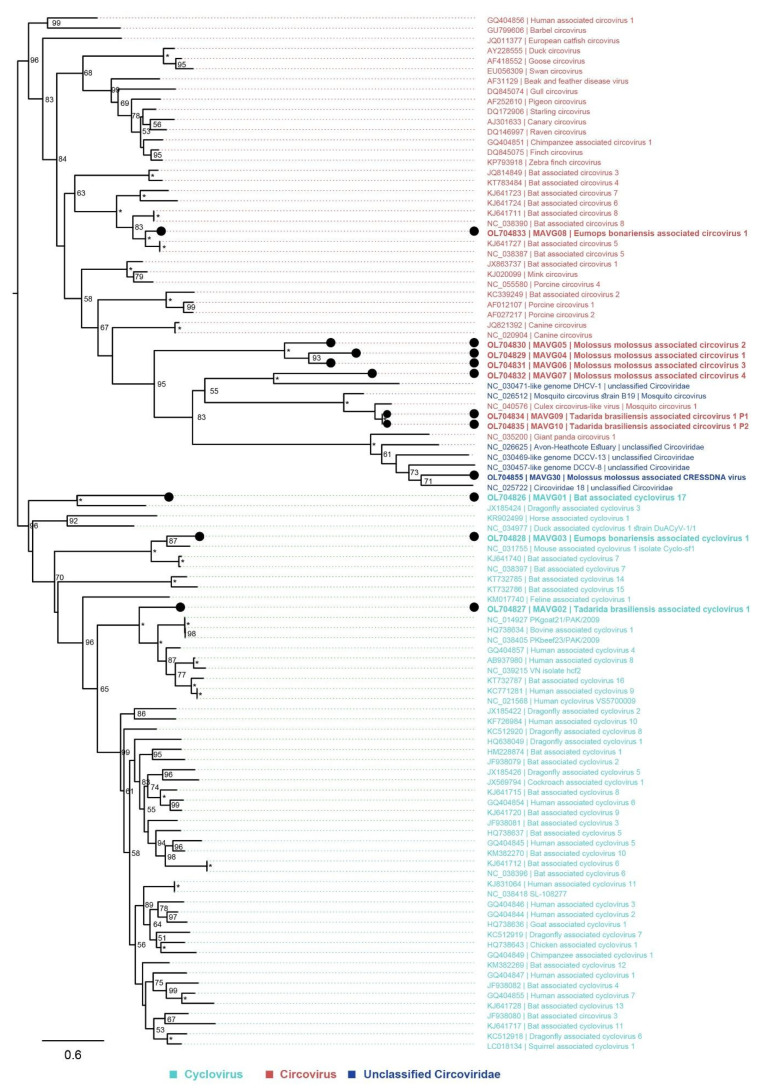
Placement of six novel circoviruses, three novel cycloviruses, and one novel circo-like virus into the phylogenetic context of other viruses clustering to the family *Circoviridae*. The phylogenetic tree was built based on the Rep protein multiple sequence alignment (aa), which was produced using mafft (v7.45) [33]. Context sequences were downloaded from the ICTV *Circoviridae* data resource (https://talk.ictvonline.org/cfs-file/__key/communityserver-wikis-components-files/00-00-00-00-83/OSD.Cir.Fig1A.Cyclovirus_5F00_circovirus_5F00_reps_5F00_aln_5F00_ed.fas, 25 September 2021). In addition, the first three most similar RefSeq sequences according to blastn searches not yet present among the primary ICTV sequences were added to the database of context sequences (*n* = 90). Newly identified viruses are marked with black dots, and genera are color-coded. Saturated node support values are shown with asterisks (*). The phylogenetic tree was constructed using iqtree v1.6 [35] with 1000 UFBootstrap replicates [37] and the phylogenetic model LG+F+R9. Tree visualization was facilitated using Figtree v1.4.4 (https://github.com/rambaut/figtree.git, accessed on 25 September 2021), and the tree was rooted in such a way that the two genera, *Circovirus* and *Cyclovirus*, were monophyletic.

**Figure 4 microorganisms-10-00266-f004:**
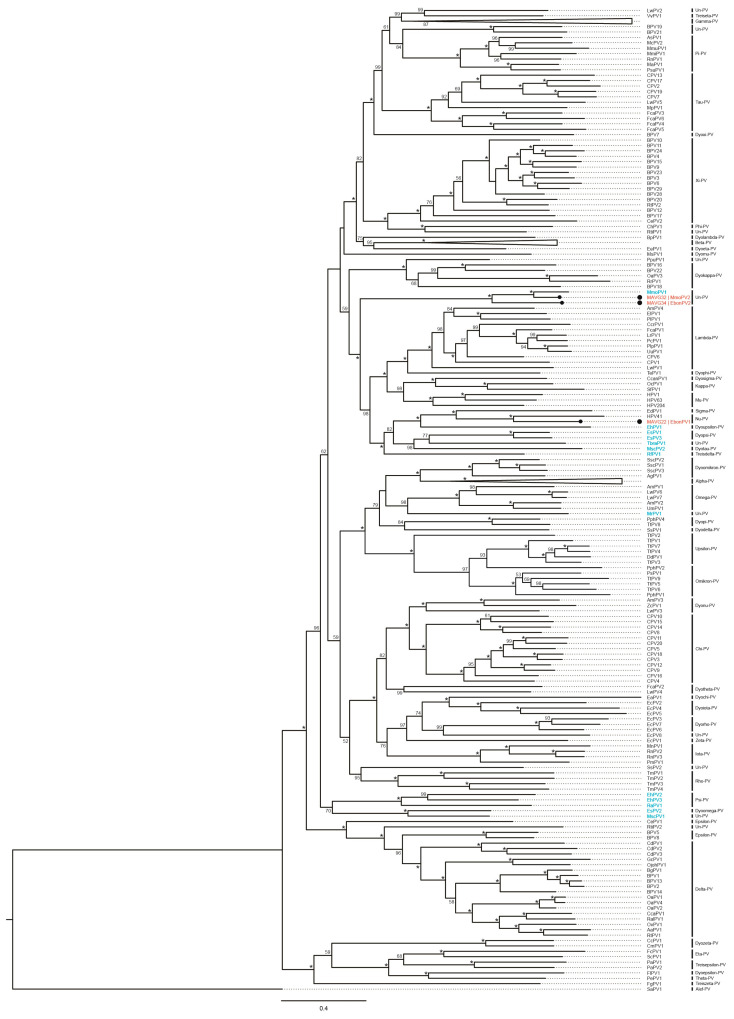
Placement of the three novel papillomaviruses (PVs) into the phylogenetic context of other viruses clustering to the family *Papillomaviridae*. The tree was built using the concatenation of the *E1, E2, L2*, and *L1* gene sequences (nt) of 384 reference PV genomes and the corresponding genes from potentially novel PVs. The phylogenetic tree was constructed using iqtree v1.6 [35] with the GTR+F+R10 model, which was chosen as the best-fitting model according to the Bayesian information criterion using ModelFinder [36]. Tree visualization was facilitated using Figtree v1.4.4 (https:/github.com/rambaut/figtree.git, accessed on 25 September 2021) and rooted at SaPV1. Branches were annotated with UF bootstrap support (1000 replicates) values [37]. Node support values < 50 are not shown, and saturated node support values are shown with asterisks (*). *Alpha*-, *Beta*-, and *Gammapapillomavirus* genera were collapsed. Novel PV types, MmoPV2, EbonPV1, and EbonPV2, are marked with black dots. Bat PV types are depicted in blue. Un-PV = unclassified PV genera.

**Figure 5 microorganisms-10-00266-f005:**
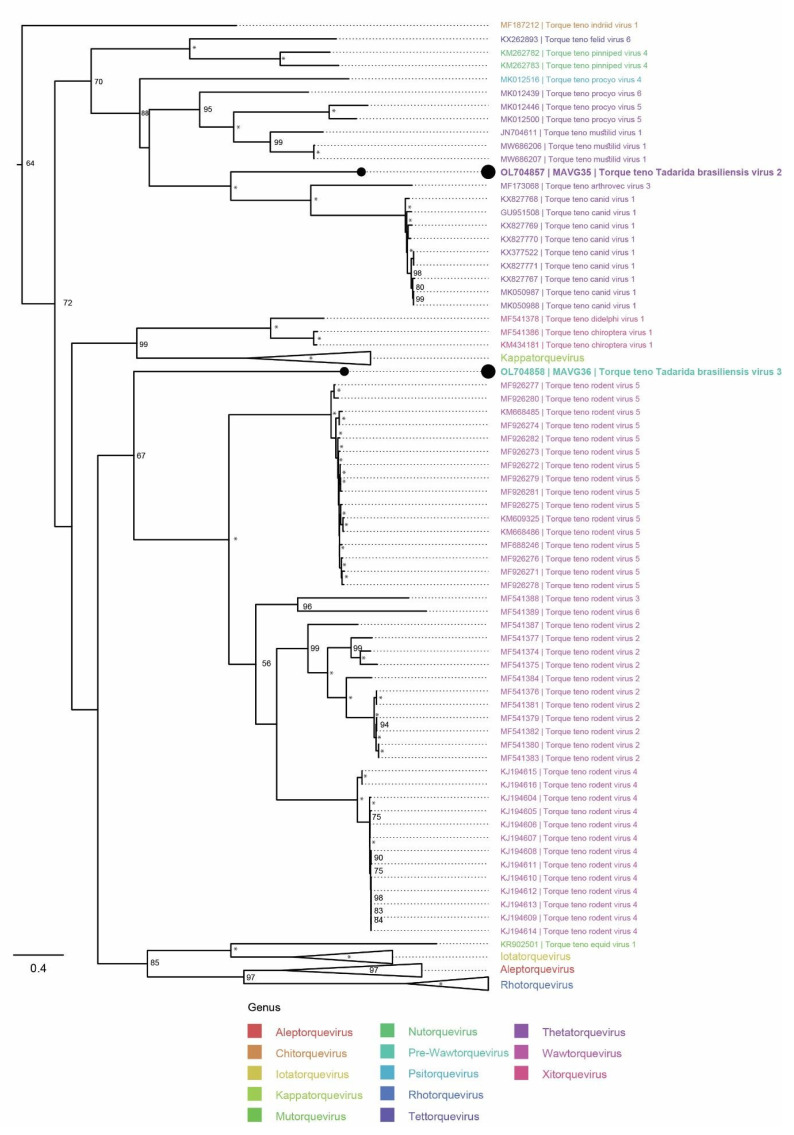
Placement of two novel anelloviruses into the phylogenetic context of other viruses clustering to the family *Anelloviridae*. The phylogenetic tree was built using the *ORF1* multiple nucleotide sequence alignment (nt), which was produced using mafft (v7.453) [33]. The anellovirus sequences evaluated in the most recent taxonomic update [29] were downloaded from GenBank and used as context (*n* = 987). Newly identified viruses are marked with black dots, and genera are color-coded. *Iota*-, *Alep*-, *Rho*-, and *Kappatorquevirus* genera were collapsed. Saturated node support values are shown with asterisks (*). Only a subtree with the 312 most relevant context sequences, rooted using the sequence MF187212, is shown. The phylogenetic tree was constructed using iqtree v1.6 [35] with 1000 UFBootstrap replicates [37] and the phylogenetic model GTR+F+R4, which was chosen as the best-fitting model according to the Bayesian information criterion using ModelFinder [36]. Tree visualization was facilitated using Figtree v1.4.4 (https://github.com/rambaut/figtree.git, accessed on 25 September 2021).

**Figure 6 microorganisms-10-00266-f006:**
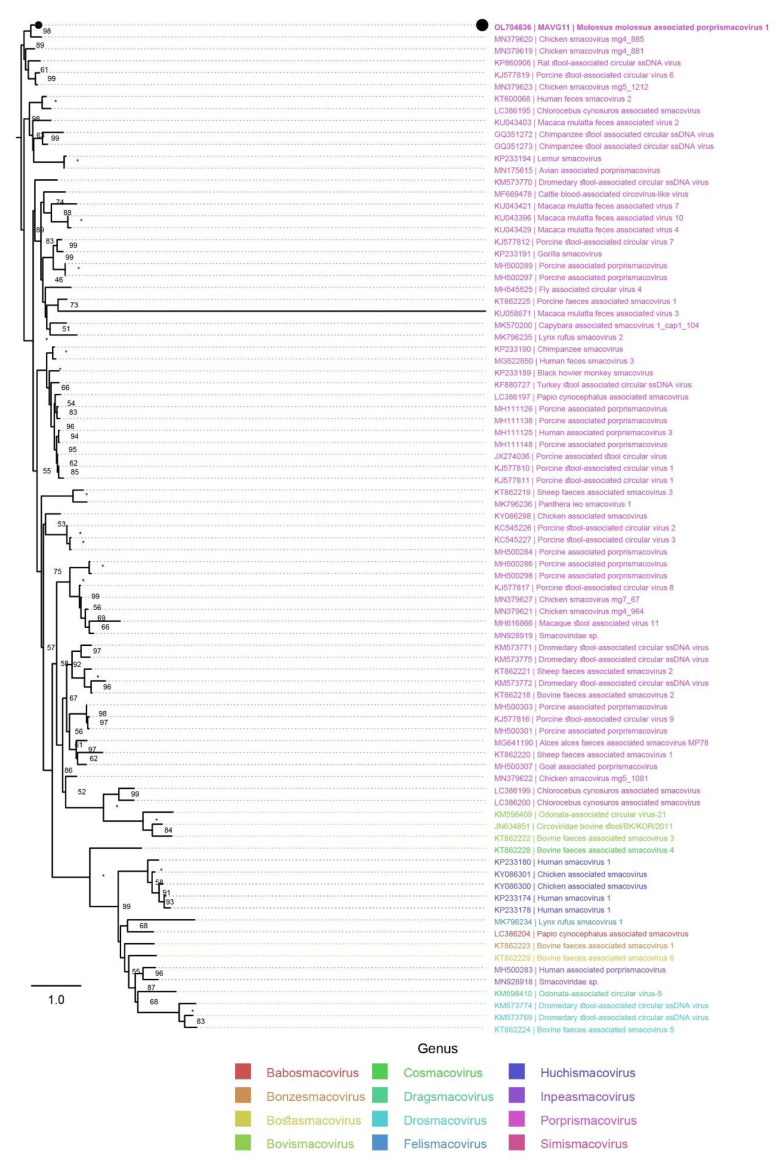
Placement of the novel smacovirus into the phylogenetic context of other viruses clustering to the family *Smacoviridae*. The phylogenetic tree was built based on the Rep protein multiple sequence alignment (aa), which was produced using mafft (v7.45) [33]. The sequences evaluated in the most recent taxonomic update of the family *Smacoviridae* [28] were downloaded from GenBank and used as context (*n* = 84). Newly identified viruses are marked with black dots, and genera are color-coded. Saturated node support values are shown with asterisks (*). The phylogenetic tree was constructed using iqtree v1.6 [35] with 1,000 UFBootstrap replicates [37] and the phylogenetic model LG+F+R9. Tree visualization was facilitated using Figtree v1.4.4 (https://github.com/rambaut/figtree.git, accessed on 25 September 2021), and the tree was rooted in a way that allowed most of the genera to be monophyletic.

**Figure 7 microorganisms-10-00266-f007:**
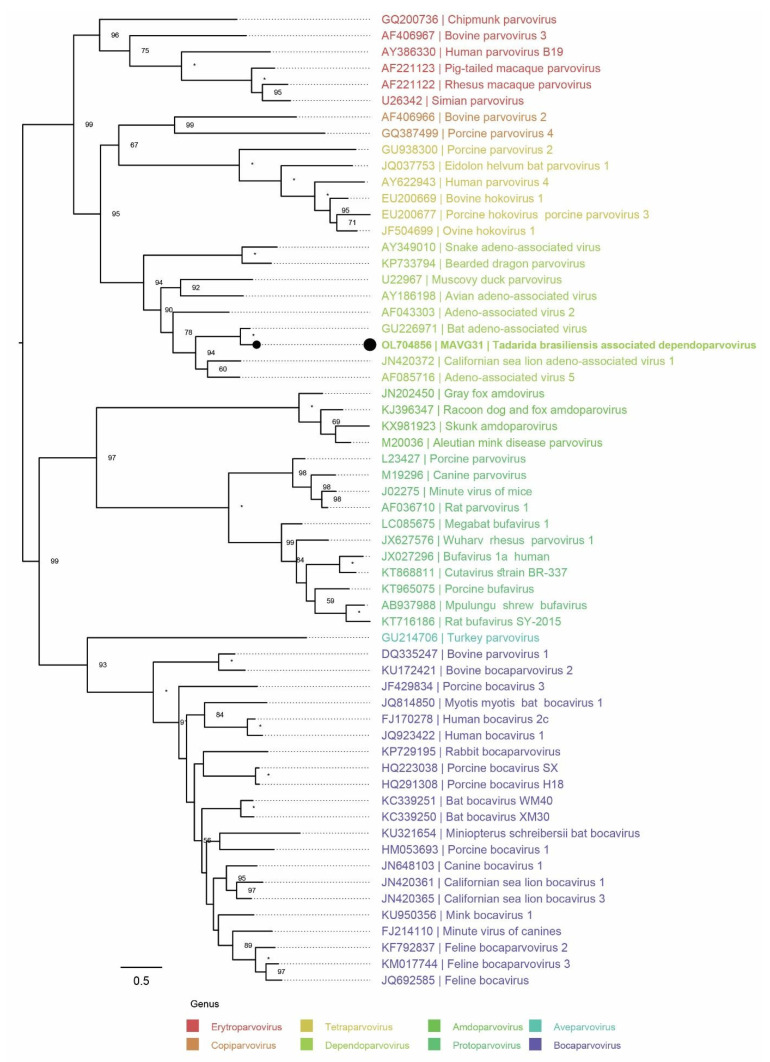
Placement of the novel parvovirus into the phylogenetic context of other viruses clustering to the subfamily *Parvovirinae*. The phylogenetic tree was built using the NS1 protein multiple sequence alignment; as suggested by the most recent relevant taxonomic update [47], the Rep40 protein subsequence was used in the case of the novel parvovirus (aa). The multiple sequence alignment was produced using mafft (v7.453) [33]. Context sequences were downloaded from the ICTV official resource page for the subfamily *Parvovirinae* (https://talk.ictvonline.org/ictv-reports/ictv_online_report/ssdna-viruses/w/parvoviridae/1055/resources-parvoviridae; 25 September 2021; *n* = 59). The black dot marks the newly identified virus. Saturated node support values are shown with asterisks (*). The phylogenetic tree was constructed using iqtree v1.6 [35] with 1000 UFBootstrap replicates [37] and the phylogenetic model rtREV+F+I+G4, which was chosen as the best-fitting model according to the Bayesian information criterion using ModelFinder [36]. Tree visualization was facilitated using Figtree v1.4.4 (https://github.com/rambaut/figtree.git, accessed on 25 September 2021), and the tree was rooted at midpoint.

**Table 1 microorganisms-10-00266-t001:** Bat samples included in each of the six sequenced sample pools by bat species, age, and collection site.

Source	Location	Collection Date	Pool ID	Sample ID	Bat Species	Age
Bat colony	Rosario	25 January 2017	1	M51	*Tadarida brasiliensis*	Young/adult
		25 January 2017		M52		
		25 January 2017		M53		
		25 January 2017		M54		
		25 January 2017		M55		
		25 January 2017	2	M56	*Tadarida brasiliensis*	Young/adult
		25 January 2017		M57		
		25 January 2017		M58		
		25 January 2017		M59		
		25 January 2017		M60		
Individual bats	Villarino Park in Zavalla	3 February 2017	4	M66	*Eumops patagonicus*	Adult
		3 February 2017		M68		
		3 February 2017		M73		
		3 February 2017		M74	*Eptesicus diminutus*	
		3 February 2017		M75		
		17 March 2017	6	M76	*Molossus molossus*	Adult
		17 March 2017		M77		
		17 March 2017		M78		
		17 March 2017		M79		
		12 April 2017	9	M90	*Molossus molossus*	Adult
		13 April 2017		M93		
		13 April 2017		M95		
		13 April 2017		M96		
		13 April 2017		M103		
		12 April 2017	11	M87	*Eumops bonariensis*	Adult
		12 April 2017		M92		
		13 April 2017		M97		
		13 April 2017		M99		
		13 April 2017		M100		

**Table 2 microorganisms-10-00266-t002:** Numbers of read pairs and contigs obtained by metagenomic workflow analysis of fecal samples of five bat species from Argentina.

Pool ID	Raw Read Pairs, *n*	Read Pairs after Quality Filtering and Trimming, *n*	Read Pairs after Host Subtraction, *n*	Read Pairs after Subtraction of Bacterial Reads, *n*	Viral Read Pairs, *n*	Viral Read Pairs that Remapped to Viral Contigs, *n*	Viral Read Pairs that Remapped to Viral Contigs, %(*)	Contigs (>500 bp), *n*	Viral Contigs (>500 bp), *n*	Viral Contigs, *%*^(^**^)^
1	991,903	716,256	711,479	591,522	92,981	92,859	15.7	797	19	2.38
2	880,008	784,591	784,248	777,013	487,928	487,872	62.8	303	16	5.28
4	736,734	641,183	641,088	587,045	37,660	37,399	6.37	8001	279	3.49
6	38,989	336,497	336,174	279,277	46,235	46,180	16.5	1910	115	6.02
9	753,848	648,621	628,771	589,624	42,214	41,876	7.10	5087	153	3.01
11	767,987	622,945	621,915	606,187	18,130	18,003	2.97	1565	109	6.97
Total	4,520,370	3,750,093	3,723,675	3,430,668	725,148	724,189	21.1	17,663	691	3.91

P1 and P2: *T. brasiliensis*; P4: *E. diminutus* and *E. patagonicus*; P6 and P9: *M. molossus*; P11: *E. bonariensis*. (*) Percentage of “Viral read pairs that remapped to viral contigs” with respect to “Read pairs after subtraction of bacterial reads”. (**) Percentage of “Viral contigs (>500 bp)” with respect to “Contigs (>500 bp)”.

**Table 3 microorganisms-10-00266-t003:** Rényi’s entropy indexes for different values of α (0–2) of fecal samples of five bat species from Argentina.

	P1	P2	P4	P6	P9	P11
H_0_	1.085	1.004	1.847	1.928	1.810	1.468
H_0.25_	1.028	1.001	1.278	1.376	1.293	1.145
H_0.5_	1.946	1.792	3.664	2.833	3.136	2.890
H_0.75_	0.914	0.444	2.674	1.938	2.222	2.002
H_1_	0.361	0.068	1.684	1.295	1.424	1.190
H_2_	0.156	0.012	0.984	0.897	0.890	0.656

P1 and P2: *T. brasiliensis*; P4: *E. diminutus* and *E. patagonicus*; P6 and P9: *M. molossus*; P11: *E. bonariensis*.

**Table 4 microorganisms-10-00266-t004:** Bray–Curtis dissimilarity of fecal samples from five bat species from Argentina.

	P1	P2	P4	P6	P9
P2	0.684				
P4	0.964	0.995			
P6	0.983	0.999	0.153		
P9	0.967	0.996	0.125	0.155	
P11	0.977	0.999	0.369	0.461	0.419

P1 and P2: *T. brasiliensis*; P4: *E. diminutus* and *E. patagonicus*; P6 and P9: *M. molossus*; P11: *E. bonariensis*.

**Table 5 microorganisms-10-00266-t005:** Novel bat-associated DNA viruses identified in fecal samples of five bat species from Argentina.

Host	Viral Family	Genus	Novel Viral Species (Yes/no ^##^; Closest Relative Genbank Acc. No. (Species); Sequence Identity ^#^)	Virus Name	MAVG	Sequencing Pool	Genbank Accession Number
*Eumops bonariensis*	*Circoviridae*	*Circovirus*	Yes; KJ641727 (*Bat associated circovirus 5*); 71%	Eumops bonariensis associated circovirus 1	MAVG08	P11	OL704833
		*Cyclovirus*	Yes; KJ641740 (*Bat associated cyclovirus 7*); 59%	Eumops bonariensis associated cyclovirus 1	MAVG03	P11	OL704828
	*Genomoviridae*	*Gemycircularvirus*	Yes; MK483082 (*Gemycircularvirus mouti 3*); 58%	Eumops bonariensis associated gemycircularvirus 4	MAVG23	P11	OL704848
		*Gemykibivirus*	Yes; KT363839 (*Gemykibivirus humas 4*); 63%	Eumops bonariensis associated gemykibivirus 1 *	MAVG24	P11	OL704849
	*Papillomaviridae*	*Nupapillomavirus*	Possible; NC_001354.1 (*Human papillomavirus 41*); 70%	Eumops bonariensis papillomavirus type 1	MAVG33	P11	OL704824
		Unclassified *Papillomaviridae*	Possible; KX812447 (*Molossus molossus papillomavirus 1*); 71%	Eumops bonariensis papillomavirus type 2	MAVG34	P11	OL704825
*Eumops patagonicus* or *Eptesicus diminutus*	*Circoviridae*	*Cyclovirus*	Yes; AB937980 (*Human associated cyclovirus 8*); 53%	Bat associated cyclovirus 17	MAVG01	P4	OL704826
	*Genomoviridae*	*Gemycircularvirus*	Yes; KF371637 (*Gemycircularvirus geras 2*); 60%	Bat associated gemycircularvirus 1 **	MAVG27	P4	OL704852
		*Gemykibivirus*	Yes; MK032742 (*Gemykibivirus minti 1*); 57%	Bat associated gemykibivirus 2	MAVG28	P4	OL704853
		*Gemykroznavirus*	Yes; MK483082 (*Gemykronzavirus hydro 1*); 72%	Bat associated gemykronzavirus 2	MAVG29	P4	OL704854
*Molossus molossus*	*Circoviridae*	*Circovirus*	Yes; JQ011377 (*European catfish circovirus*); 47%	Molossus molossus associated circovirus 1	MAVG04	P9	OL704829
			Yes; JQ011377 (*European catfish circovirus*); 47%	Molossus molossus associated circovirus 2	MAVG05	P9	OL704830
			Yes; JQ011377 (*European catfish circovirus*); 48%	Molossus molossus associated circovirus 3	MAVG06	P9	OL704831
			Yes; KJ641727 (*Bat associated circovirus 5*); 48%	Molossus molossus associated circovirus 4	MAVG07	P9	OL704832
	*Genomoviridae*	*Gemycircularvirus*	No; MT138090 (*Genomoviridae* sp. isolate wftbif32cir1); 88%	Molossus molossus associated gemycircularvirus 3	MAVG15	P9	OL704840
			Possible; MH047857 (*Gemycircularvirus mocha 1*); 77%	Molossus molossus associated gemycircularvirus 1	MAVG13	P9	OL704838
			Yes; KY308268 (*Gemycircularvirus echiam 1*); 58%	Molossus molossus associated gemycircularvirus 2	MAVG14	P9	OL704839
			Yes; KT862242 (*Gemycircularvirus chicas 2*); 60%	Molossus molossus associated gemycircularvirus 5 **	MAVG25	P6	OL704850
			Yes; KF371637 (*Gemycircularvirus geras 2*); 72%	Molossus molossus associated gemycircularvirus 4	MAVG19	P9	OL704844
		*Gemygorvirus*	Yes; MH939362 (*Gemygorvirus poaspe 1*); 59%	Molossus molossus associated gemygorvirus 1	MAVG20	P9	OL704845
		*Gemykibivirus*	Yes; MK032742 (*Gemykibivirus minti 1*); 64%	Molossus molossus associated gemykibivirus 1 *	MAVG12	P9	OL704837
			Yes; MK032742 (*Gemykibivirus minti 1*); 62%	Molossus molossus associated gemykibivirus 2	MAVG16	P9	OL704841
			Yes; 63% similar to MK032742 (*Gemykibivirus minti 1*)	Molossus molossus associated gemykibivirus 3	MAVG17	P9	OL704842
			Yes; KT363839 (*Gemykibivirus humas 4*); 61%	Molossus molossus associated gemykibivirus 4	MAVG18	P9	OL704843
			Yes; MK032742 (*Gemykibivirus minti 1*); 64%	Molossus molossus associated gemykibivirus 5	MAVG21	P9	OL704846
			Yes; KJ547642 (*Gemykibivirus sewopo 2*); 62%	Molossus molossus associated gemykibivirus 6	MAVG22	P9	OL704847
		*Gemykroznavirus*	Yes; MK483082 (*Gemykronzavirus hydro 1*); 64%	Molossus molossus associated gemykronzavirus 1	MAVG26	P6	OL704851
	*Circoviridae*	Unclassified *Circoviridae*	Yes; MN582084 (CRESS virus sp. ctf7a5, complete genome); 58%	Molossus molossus associated CRESSDNA virus	MAVG30	P9	OL704855
	*Smacoviridae*	*Porprismacovirus*	Yes; KP860907 (*Porprismacovirus ratas 1*); 67%	Molossus molossus associated porprismacovirus 1	MAVG11	P9	OL704836
	*Papillomaviridae*	Unclassified *Papillomaviridae*	No; KX812447 (*Molossus molossus papillomavirus 1*); 80%	Molossus molossus papillomavirus type 2	MAVG32	P9	OL704823
*Tadarida brasiliensis*	*Parvoviridae*	*Dependoparvovirus*	Yes; GU226971 (*Bat adeno-associated virus*); 80%	Tadarida brasiliensis associated dependoparvovirus	MAVG31	P1	OL704856
	*Circoviridae*	*Cyclovirus*	Yes; HQ738634 (*Bovine associated cyclovirus 1*); 62%	Tadarida brasiliensis associated cyclovirus 1	MAVG02	P2	OL704827
		*Circovirus*	Yes; NC_040576 (*Culex circovirus-like virus*); 75%	Tadarida brasiliensis associated circovirus 1	MAVG09, MAVG10	P1, P2	OL704834, OL704835
	*Anelloviridae*	*Thetatorquevirus*	Yes; MT010529 (unclassified canid anellovirus); 64%	Torque teno Tadarida brasiliensis virus 2	MAVG35	P2	OL704857
		Unclassified *Anelloviridae*	Yes; HM633238 (*Torque teno sus virus k2a*); 55%	Torque teno Tadarida brasiliensis virus 3	MAVG36	P2	OL704858
Total	7	13	31	35	36	6	

* and ** indicate that the virus belongs to the same viral species as another virus that was identified during this study. ^#^ Sequence identities were measured using SDT [32] at the genomic level relevant for species demarcation as prescribed by the ICTV: complete genome identities for *Circo-*, *Smaco-* and *Genomoviridae* and *L1* and *ORF1* gene nucleotide sequence identities for *Papilloma-* and *Anelloviridae*, respectively, and the amino acid sequence identity of NS1 for *Parvoviridae*. ^##^ Proposition. ICTV handles the official taxonomical decisions regarding viral species definition and naming.

## Data Availability

The novel viruses reported in this article are openly available in the GenBank/EMBL/DDBJ database with the following accession numbers: OL704823–OL704858. The relevant raw high throughput sequencing data obtained in this study were deposited at the NCBI Sequence Read Archives (SRA) under BioProject ID PRJNA786972.

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
