# Peer review of "Viral Metagenomic Data Analyses of Five New World Bat Species from Argentina: Identification of 35 Novel DNA Viruses"

_microorganisms, 2022, doi:10.3390/microorganisms10020266_

Round 1

Reviewer 1 Report

Bolatti et al. investigated the diversity of DNA viruses in fecal material from 5 species of New World bats from two different locations in Argentina. Overall, 29 samples collected into 6 pools were tested. The authors first analyzed the overall viral diversity and compared viral diversity between pools and locations and observed marked differences between samples. In particular, two pools including samples collected from one specific species and specific location were characterized by a lower diversity and this could be due to the viral species or to the fact that those bats live in a more urban area. Additionally, the authors take a close look at long viral contigs that represent (near) complete viral genomes and investigate their phylogenetic relationships with other members of the families they belong to. They find 18 genomoviruses, 11 circoviruses, 1 smacovirus, 1 parvovirus, 2 papillomaviruses and 2 anelloviruses. The study is well performed, and sequences have been thoroughly analyzed (phylogeny + genome organization). Although highly speculative, conclusions are supported by results and other studies in literature. The manuscript is well written. I have, therefore, only minor suggestions for the authors.

  1. My only bigger concern is virus nomenclature. While it is perfectly acceptable (also encouraged) to give names to identified viruses, I would refrain to assign specific taxonomic designation (species name) to them. This is because species names are decided by the ICTV, after evaluating whether a species is eligible for being included in the taxonomy, and these names follow specific rules and also publication order. It is not advisable to assign species names in publications because it can happen (and it often does) that various authors give the same name to different species, and this creates chaos afterwards. For example, there are at least 4 different “equid herpesvirus 7”, which are different viruses found in different hosts, that were named in the same way in publications, independently from the coordination of the ICTV. Additionally, the ICTV is moving towards a binomial nomenclature system for species and, while the taxonomy of some families has already been updated (e.g. Genomoviridae), the taxonomy of many viral families still has to be revised (e.g. Parvoviridae, Anelloviridae, Circoviridae) and all species names will change within 1-2 years. Assigning (even proposing) an official species name is a bad idea as it might create confusion in the future.
  2. In the text you refer to the viruses using the MAVG IDs, but those are not indicated in the trees. Therefore, to understand what the text is referring to, one has to go back to the table, find the ID, figure out how the virus was named, and then go back to the tree. I strongly advise to include sequence IDs in the trees too to make everything clearer.
  3. Figures are all different from one another. In some figures you used both species and common names, in others you used only species or only common names and in the PV tree you used abbreviations. Additionally, in all but one trees the sequences of this study are indicated by a dot. I think it would be better if figures were made homogeneous.
  4. I would add something in the intro about the bat species you investigated. E.g., were they all insect-eating? What kind of “close contacts” do they have with humans in Argentina? Are they locally distributed or do they have a wide geographic range?
  5. To make Figure 1 clearer, I would indicate in panel A the host of the viral families indicated (they are already grouped by host anyway, you can simply make some horizontal lines to improve clarity).
  6. Table 3 and Figure 2 say the exact same thing and you don’t need both. I suggest moving one of the 2 in the supplementary material.
  7. Table 5 is unreadable! You should stretch it over the whole length of the page. Also, in the 4th column you give some identity values, but it is nowhere specified how they were calculated. As each viral family has different rules for species demarcation, I am assuming that those values reflect those rules (e.g. NS1 protein for parvoviruses or nucleotide complete genome for circoviruses). If these rules were not followed in the table, they should be. In any case, considered region and whether aa or nt were used should be indicated (you could use footnotes).

Other

- Official approved species names (e.g. “Torque teno chiroptera virus 1”) should be written in italics, while viral common names (e.g. “Bat adeno-associated virus”) should not be italicized. This rule must be followed and applies to section titles and figures too.

- Line 35. You only looked at DNA viruses, so I suggest to replace “viral diversity” with “diversity of DNA viruses”.

- Lines 101-2. You forgot the italics.

- Table 1 should be moved where it is first referenced.

- Lines 272-3. You forgot the italics.

- Line 326. There is a title formatting error.

- In the caption of all trees you have kept the doi link as well as the bibliographic reference for used methods. I would remove the links.

- Line 527. Parvoviruses do not have a “characteristic 3′-terminal polyA nucleotide signal”, they have inverted terminal palindromes.

- Line 555. This sentence seems to indicate that all viruses that are highly pathogenic to humans come form bats. Please rephrase this sentence.

- Lines 556-8. The grammar in this sentence is a bit weird. Please, rephrase.

- Line 610. “found”, not “founded”.

- Lines 628-9. This is not strong evidence for bats being the actual host of these viruses. Please, rephrase.

- Lines 692-3. The association between TTVs and immune status was evaluated for viruses in blood, and it may be very different in this case. I would remove this sentence because it is too speculative.

- Lines 733-4 and 747-9. I am not sure how the connection between reproduction cycle and virus shedding was made. Do you have elements to make these statements? These are highly speculative statements and I suggest removing them.

Reviewer 2 Report

In the research article titled "Viral metagenomic data analyses of five New World bat species 2 from Argentina: identification of 35 novel DNA viruses", Bolatti EM et al. applied shotgun mtegenomics analysis to investigate fecal samples of 29 individuals from five species living at two different geographical locations. In the study 35 novel DNA viruses were identified and reported.

Overall, the finding is importent and is should be sufficient to publish in the journal of Microorganisms.

I have only a few minor comments:

1) In Table 2, how the "Viral read pairs mapped to viral contigs%" was calculated? If  the number under "Viral read pairs mapped to viral contigs" was divided by the number under "Viral read pairs", the percentage is close to 100%. For example, for Pool ID 1, 92859/92981*100=99.87%, not the reported 15.7%.

2) Figure 1A should be corrected to use relative abundance for generation of heatmaps. Absolute numbers are not comparable across samples, since the sequencing depth varied.

3) In the phylogenetic trees, it will be helpful to label the newly identified virus by host and/or geographical location.

Reviewer 3 Report

Bolatti et al have presented an interesting study of an area that is in great need of further study, this being the viromes of new world bat species. The manuscript is well presented and the writing is compelling, though I find the results section to be a bit long and could use cutting down to just the most important findings.

I have a few minor comments and suggestions: 

-Why were human reads subtracted? Surely there should not be any human reads in this sample and if there is, this suggests contamination which would throw all results into question. And surely given the DNase digestion undertaken prior to sequencing, there should be minimal host reads remaining.

Figure 1B: This figure is confusing and unnecessary

Figure 1C: do you mean the virus host kingdom?

Table 5 is difficult to read. To simplify the table I would recommend removing column 4 (“novel virus species”) and instead have a column labelled ‘top blast hit’ and one labelled ‘similarity to top blast hit’ and specify if this similarity is at the nucleotide or amino acid level. It would be advantageous to format the table so that it spans a single page so that the reader can refer back when reading the rest of the results section.

Line 348: given that three assembly tools were used, was this assembly common across all tools?

phylogenetic trees: please make it clearer if these are amino acid or nucleotide trees.

The information about the phylogenetic classification of each family is not necessary in the results section unless it directly impacts on the research. This can be discussed in the discussion if necessary to the work but a lot of these descriptions are wordy and seem unnecessary. 

Line 595: these RNA viruses are likely endogenous viral elements

Minor corrections: 

-line 61: “accessed on 11st November 2021” should say 11th.

Line 114: “no animals were sacrificed” I think a better way of saying this would be “no animals were harmed or required euthanasia”

Line 692: "they might represent bat normal viral flora" - correction 'they might represent a bats normal viral flora' ? 

line 738: I don't think anthropomorphism is the word you mean here- this means an object with human like characteristics. I would use the term urbanisation 
